# Smartphone-Based Cost-Effective Pavement Performance Model Development Using a Machine Learning Technique with Limited Data

**Samiulhaq Wasiq and Amir Golroo ***

Department of Civil and Environmental Engineering, Amirkabir University of Technology, 350, Hafez Ave, Valiasr Square, Tehran 1591634311, Iran; samr1@aut.ac.ir
* Correspondence: agolroo@aut.ac.ir; Tel.: +98-21-64543048; Fax: +98-21-66414213

**Abstract:** Road networks play a significant role in each country's economy, especially in countries such as Afghanistan, which is strategically located in the international transit path from Europe to East Asia. In such a country, pavement performance models are fundamental for the pavement maintenance planning that provides high-quality infrastructure for transporting goods and travelers. However, due to the lack of a budget for pavement monitoring and maintenance in Afghanistan, transportation networks and pavement condition data have not been widely acquired for the development of a pavement performance model. The main aim of this study is to use a machine learning technique to, for the first time, develop a pavement performance model for Afghanistan that uses simple, cost-effective, and fairly accurate data—collected via smartphones—and that is based on a case study of over 550 km of Afghanistan's highways. First, the current condition of Afghanistan's road network is investigated using a smartphone. Then, collected data are prepared and analyzed so as to estimate the pavement condition index (PCI). Finally, a pavement performance model for PCI is developed using pavement age with an adequate coefficient of determination of 0.70 and successfully validated. It is concluded that the proposed approach is efficient and effective when developing a performance model in other developing countries encountering such data and budget limitations.

**Keywords:** pavement performance models; machine learning; regression modelling; smartphones

## 1. Introduction

Transportation networks play important roles in each country's economy because they transfer passengers and commodities from their origins to their destinations; as a result, road authorities attempt to maintain these networks at a high level of service. For this purpose, the concept of pavement management has been developed and implemented by road authorities. From the lens of agency and user costs, researchers have attempted to scrutinize the advantages of deploying a pavement management system (PMS) in order to optimize the life cycle cost of road pavement [1].

A PMS is a primary need of each road authority. Various countries around the globe have tried to build their PMS according to their requirements, technologies, budgets, etc. Among these, different countries, e.g., the United States [2], Canada [3], India [4,5], Egypt [6], Saudi Arabia [7], Chile, Paraguay [8], Poland [9], Portugal [10], and Mexico [11], have developed tailor-made PMSs for their countries.

A PMS starts with pavement material and design, continues with construction, implementation, and maintenance, and finally ends with pavement end-of-life and recycling. The main objective of a PMS is to conduct maintenance planning in an optimal manner that both maintains the road at a high level of service and minimizes the associated costs. For this purpose, the vital requirement is a pavement performance model which can predict pavement conditions. Using the performance model, road authorities can determine when a suitable time would be for maintenance actions on the pavement before it drastically

deteriorates. Several countries have developed pavement performance models, though developing countries have been rarely successful in building such models due to budget constraints. There has not been any pavement performance model developed for Afghanistan's road network to date due to the lack of historical data and the unavailability of affordable pavement condition data collection tools with which to monitor the current condition of pavement. This study proposes a simple, applicable approach to fill this gap.

The objective of this paper is to develop a pavement performance model that is cost-effective and simple to implement while offering an adequate level of accuracy that can be applied for pavement maintenance planning and life cycle cost analysis for developing countries.

The scope of this research was to develop an asphalt pavement performance model for different families of pavement with almost the same pavement structure, traffic loads, and weather conditions, focusing on the national highways and provincial roads of Afghanistan. The pavement condition index (PCI) was applied to develop a performance model for pavement age through the application of images taken by a smartphone mounted on a probe vehicle. For this purpose, pavement distress was detected through images via a semi-automated technique, i.e., pavement distress type, severity, and density were determined to compute the PCI for monitored pavement sections. Finally, having applied univariate linear and non-linear regression techniques, a set of pavement performance models was developed, and the best-fitted model was introduced and successfully validated.

## 2. Literature Review

### 2.1. Pavement Performance Model Types

There are different pavement performance model types applied by researchers to express the condition of pavement, and these can be categorized into four types: subjective, deterministic, stochastic, and Bayesian. The subjective models subjectively evaluate pavement conditions using expert panels [12–14], while the deterministic models objectively describe pavement conditions in the future [15–17]. Moreover, stochastic models assign a probability distribution function to inputs, instead of a specific value, and arrive at such a function as an output [18–20]. Finally, the combination of initial qualitative data with quantitative data (i.e., captured from field or labs) results in increased accuracy of data via Bayesian models [21]. Generally speaking, regarding the amount of pavement condition data accessible, two strategies can be applied to develop pavement performance models. The first strategy needs limited data (short term), while the other requires much data (long term). The former strategy can build a performance model either with limited data, such as a regression technique or with at least two-time section data, such as the Markov chain method; these are deterministic and stochastic modeling techniques, respectively. The latter strategy, which includes neural networks, is a deterministic modeling technique that needs multi-year pavement condition data to model pavement performance [22,23].

### 2.2. Modeling Techniques

In order to develop an appropriate model, a number of principles should be considered, including model type and specification/simplicity, evaluation metrics, generalization and validation, engineering credibility, and model assumptions. The model type and specifications are closely related to the correlation between the dependent and independent variables. If the correlation is linear, a linear model is suggested; otherwise, a non-linear model is preferred. The correlation between dependent and independent variables can easily be recognized using the correlation coefficient (r) for linear correlation and a simple scatter plot to distinguish non-linear patterns. The simpler the model, the better and more applicable the model is, especially in the case of real usage by road authorities in developing countries.

Evaluation metrics are applied to measure the performance of the model. Generally speaking, these metrics indicate how well the model fits the data. To find the best fit, the error/residual (difference between actual data/ground truth and model prediction) should

be minimized. The most common metrics in a linear regression model are the root mean squared error (RMSE) and the determination coefficient ($R^2$). If model metrics are low, that could be a sign of model misspecification or a lack of significant independent variables in the model.

The model generalization is considered to ensure that the model can predict well the test (unseen) data. If a model overfits the train (seen) data, it could perfectly predict the train data but poorly perform on the test data. In a machine learning (ML) context, it is noted that this model memorizes the train data and cannot predict other data. Again, the model misspecification (e.g., using higher order in the polynomial model) could be the reason. On the contrary, model misspecification would happen in a case where a lower order of polynomial or less than enough independent variables is employed in the modeling which is called underfitting. The validation process that compares the predicted versus actual data can be utilized through plotting or conducting a two-sample *t*-test.

The model, in addition, should make engineering sense. Not only should it follow the related literature, but it should also express logical sense in positive or negative correlations between dependent and independent variables. For instance, pavement loading as an independent variable should have a negative correlation (i.e., the negative sign in the model) with pavement condition as a dependent variable, i.e., the higher the pavement loading, the lower the pavement condition would be.

Finally, model assumptions should be checked after model development. For instance, in linear regression modeling, residual diagnostics, multicollinearity, heteroscedasticity, and autocorrelation should be scrutinized. In short, the model residuals should, respectively, follow the normal distribution, independent variables should not highly correlate with each other, the variance of the dependent variable (or errors) should be constant, and errors should not be autocorrelated over time.

### 2.3. Pavement Performance Modeling Procedure

According to the model selection principles mentioned above and regarding the related literature, the regression model is one of the most appropriate and simple techniques employed for pavement performance modeling, especially in the case of limited available pavement condition data. In developing countries, a major challenge to be handled when modelling pavement performance is the conducting of affordable and adequately accurate pavement condition data collection when dealing with no or limited historical pavement data.

First, to develop a pavement performance model, a pavement index should be selected, such as the international roughness index (IRI) or the pavement condition index (PCI). The performance model would predict such an index over time. The index selection is mainly dependent on which characteristics of pavement are studied. For instance, if pavement surface defects are researched, the PCI would be a proper index, while IRI would be more appropriate for investigating the smoothness of pavement. Although PCI represents a measure of pavement surface conditions that considers various pavement distresses, it has been acknowledged that the IRI exhibits a stronger correlation with important pavement aspects such as quality of ride, road safety, and fuel consumption. Recent studies have extensively explored the impact of the IRI on these aspects. These studies involve the development of vehicle dynamic models, specifically a 3D car model and a 3D truck model, to predict road IRI and assess its influence on vehicle rolling resistance [24,25].

Second, the pavement condition data should be acquired. Such data can be collected manually or automatically. The former has been rarely employed by developed countries, e.g., in the United States less than 2% of states apply a manual approach to pavement data collection [26]. This approach is conducted by experts filling out forms by walking on pavements and investigating distress and is costly, dangerous, labor intensive, and time-consuming. However, the automated approach, which is carried out using automated data collection vehicles, would overcome the above-mentioned drawbacks. Operating such an approach is not, however, affordable for developing countries, due to the high

cost of purchasing, operating, and maintaining such vehicles. Recently, with the advent of new technologies, novel methods have been deployed to assess the condition of pavement by applying packages of sensors [27] or smartphones [28]. Smartphones may be the only available tools that, not only do not need extra infrastructure, but also have useful and adequately accurate embedded sensors, e.g., an accelerometer, camera, gyroscope, and geographical positioning system (GPS) with which to monitor the ambient condition. This is also affordable, attainable, and user-friendly without special training, which is especially useful for developing countries. For instance, a smartphone's camera can capture images that are deployed to detect surface distress. Furthermore, they may provide more information about the pavement condition, such as the signs of structural inadequacy (by observing pothole, alligator cracking, or rutting), safety (by detecting bleeding, raveling, or weathering), and roughness (by monitoring shoving, corrugation, depression, bump and sags or swells) [29].

Third, a performance model should be developed using the selected index and a modeling technique. Several studies have been conducted around the globe to develop a pavement performance model for regions, cities, and countries [30] that can predict pavement condition rating based on pavement type, thickness, pavement age, traffic, and the current pavement condition rating. Other researchers have developed an empirical–mechanistic-based prediction model for IRI using asphalt layer thickness, environmental conditions, subgrade strength, and the structural number [31]. Ref. [32] employed the same index to develop a performance model using an artificial neural network (ANN) and regression techniques based on the primary IRI, standard deviation of the rutting, transverse, and alligator cracks. Some investigators have applied a massive database such as that of the Long-Term Pavement Performance (LTPP) program, to build up a pavement performance model [33]. They developed six performance models specifically for wet and dry non-freeze climatic zones for single indicators of distress such as alligator, longitudinal, and transverse cracking, raveling, bleeding, and rut depth. Younos et al. utilized the PCI as an index to develop a performance model based on pavement age, pavement cracks, traffic loading, and maintenance effects using LTTP data through artificial intelligence and Markov chain techniques [34]. Gao et al. investigated the effect of the ambient environment, i.e., variations of weather, along with seasonal and annual changes on the pavement performance at a network level [35].

However, such an attempt has rarely been conducted for developing countries with little or no historical and current pavement condition data. The lack of performance models for such countries has resulted in an inadequacy of information for proactive pavement maintenance planning. Most of the time, developing countries evaluate the current condition of pavement and employ this evaluation for reactive maintenance planning, such as in the cases of the evaluation of the PCI in Yaman [36], monitoring of pavement condition in Kabul, Afghanistan [37], assessing road roughness in India [38], and investigation of the IRI in New Mexico [39]. Nonetheless, few developing countries have attempted to develop performance models for their cities or countries. For instance, Semnarshad et al. developed a pavement performance model using the weighted summation of the PCI, IRI, and central deflection of falling weight deflectometer via analytical hierarchy analysis for Iran [40]. Ahmed et al., deployed pavement surface distress data, including cracking, pothole, bleeding, rutting, patching, and depression, from 1100 sections in Baghdad, Iraq to develop a pavement performance model using the stepwise regression technique [41]. Tchemou et al. developed a degradation model for Cameron based on rutting depth measured in the field. They employed a finite element program to model the degradation using permanent stress and deformation causing the rutting [42]. All mentioned research studies tried to predict a pavement index based on single or multiple variables via a modeling technique defined based on available data, budget constraints, and a pavement index that they aimed to assess.

To sum up, with around 19,327 km of Afghanistan's national highways, provincial, and district roads (Table 1), and a variety of weather conditions [43], no specific pavement

performance models have been built for Afghanistan's roads. Only few pavement condition monitoring attempts have, to date, been carried out to prescribe reactive pavement maintenance actions for road conditions in Afghanistan [44]. Thus, the main research objective was to apply a cost-effective data collection tool to develop a performance model of Afghanistan's road network through the application of the PCI and employing an appropriate regression modeling technique.

**Table 1.** Selected list of Afghanistan's road network.

| Road Class | Asphalt | | Gravel | | Earthen | | Total | |
|---|---|---|---|---|---|---|---|---|
| | km | % | km | % | km | % | km | % |
| National highways | 4598 | 59 | 2042 | 20 | 214 | 20 | 6854 | 35 |
| Provincial Roads | 813 | 10 | 464 | 4 | 117 | 11 | 1394 | 7 |
| District Roads | 2482 | 31 | 7853 | 76 | 744 | 69 | 11,079 | 58 |
| Total | 7893 | 40 | 10,359 | 54 | 1075 | 6 | 19,327 | 100 |

## 3. Research Methodology

After a thorough literature review, a pavement condition index, data type, and sensor selection were carried out. Then, the design of an experiment was conducted for pavement condition data collection. After data collection, data preparation was performed to clean the data. Afterward, the data were analyzed (i.e., pavement distress was detected and measured) to estimate the PCI. Then, various models were developed via different regression techniques to predict PCI based on pavement age. Finally, the best model was determined and validated to ensure that the model was perfectly generalized. The research methodology is elaborated in Figure 1.

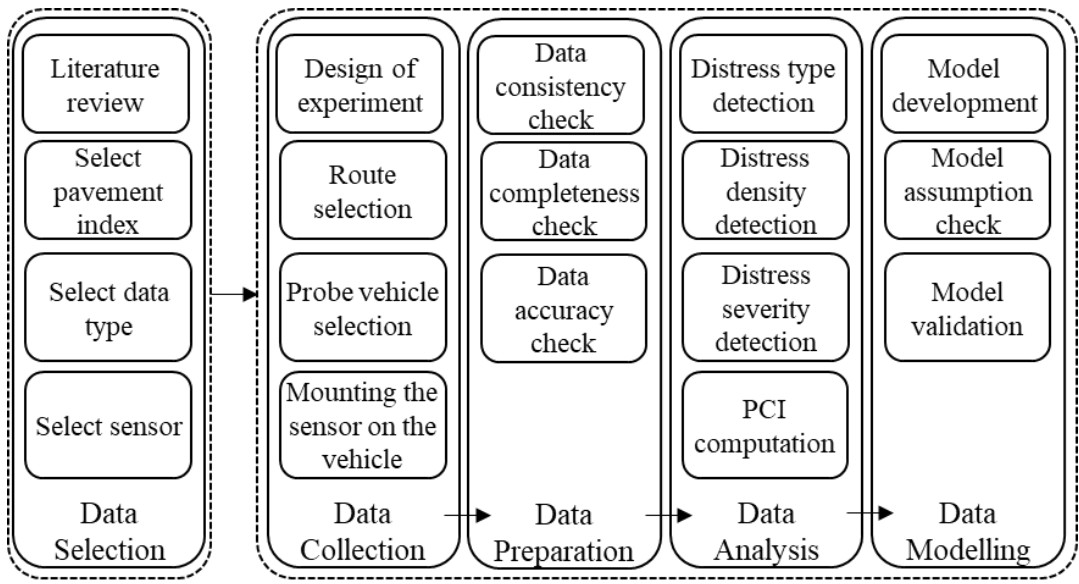

**Figure 1.** Research methodology.

### 3.1. Data Selection

The pavement index which was selected in this research was the PCI because it can not only capture the pavement surface distress, but also reflects structural adequacy, pavement roughness, and road safety. In terms of structural adequacy, some defects, such as alligator cracking or rutting, would alarm the structural inadequacy. Moreover, vertical deformation distresses such as corrugation, bumps and sags, shoving, and swell would reflect pavement roughness. Finally, potholes, raveling, weathering, and bleeding may represent a lower level of road safety.

The data type that was applied herein to compute the PCI was images captured from the pavement. Through the application of images, according to ASTM 6433, all 20 asphalt pavement distress types were detected, and their severity and density were introduced [45]. Other sources of data, such as the pavement profile or macro-texture (using laser or lidar), could enhance the quality of the data; however, these were not applied herein due to budget constraints.

The sensor that was deployed in the data collection process was a smartphone camera. The camera could acquire pavement images continuously and with a high quality. There are other embedded sensors in the smartphone that were utilized in the data collection in this study along with the camera. For instance, there was an image stabilizer sensor which assisted in capturing clearer images. Additionally, the GPS was applied to geo-referenced all captured images. The other available sensor embedded in smartphones is an accelerometer that can capture the vehicle vibration, a so-called a vibration-based method, leading to the estimation of road roughness. Although such a sensor can provide some information about road roughness, it suffers from some shortcomings, i.e., the outcome of the smartphone's accelerometer highly depends on the data collection specifications, such as the accuracy of the smartphone's accelerometer (data collection frequency), the position of the smartphone in the vehicle and how it is affixed in the vehicle, the vehicle's suspension system, the vehicle speed, and the smartphone model. However, the smartphone image-based data collection method does not depend on these specifications. The only concern is to fix the smartphone in the vehicle in a position where it can capture the pavement defect clearly, which can be achieved with a holder mounted on the top of the windshield. Due to the advantages of the image-based smartphone data collection method over the vibration-based method, the former approach was applied in this research.

### 3.2. Data Collection

For the design of the experiment, a route of about 558.7 km, mostly located in the southern part of the Afghanistan ring road, was selected given the fact that it encompassed all pavement families (discussed below) with approximately equal lengths, as depicted in Figure 2.

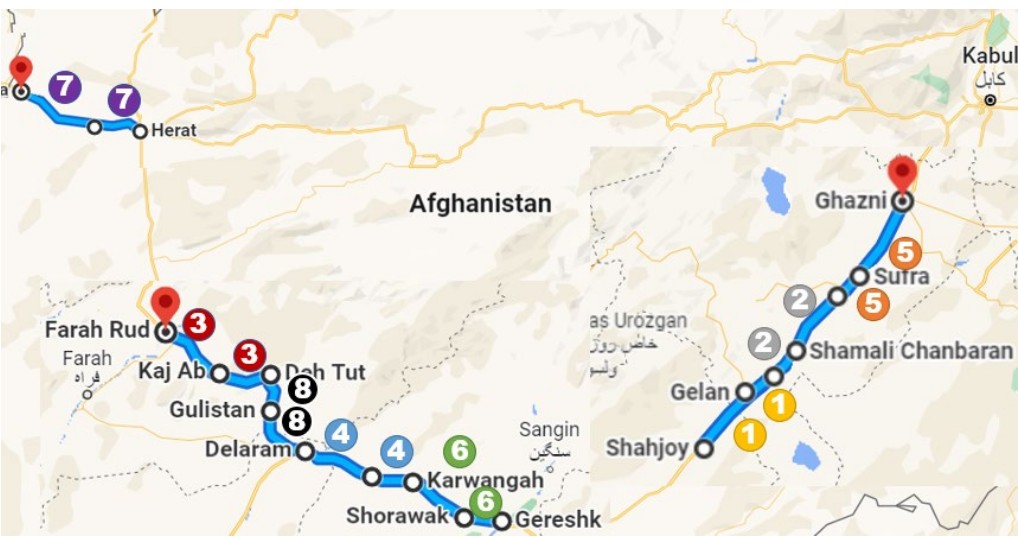

**Figure 2.** Afghanistan's monitored road network (numbers presents pavement families).

The data, which comprised pavement videos, were captured via a built-in application using a smartphone (Samsung Galaxy M32) mounted inside of a car on the windshield behind the mirror as shown in Figure 3a. The data collection was conducted one time by an individual driver using a sedan car mostly, driving at constant speed between 50 and 70 km/h (speed variety did not cause any error) according to the pavement

condition, over three days in February 2021 in mornings and afternoons with enough sunlight, i.e., one hour after and before sunrise and sunset, respectively, in dry conditions. Before data collection, several dry runs were conducted to ensure the best location for mounting the smartphone in the car, its setting, and the quality and validity of captured videos. The videos were split into images at a constant distance of 5 m to ensure that more than 50% overlap existed between two consecutive images so as to capture all pavement distresses. Different pavement distresses were detected from the images, including alligator cracking, weathering, rutting, raveling, corrugation, and potholes, as shown in Figure 3b–f.

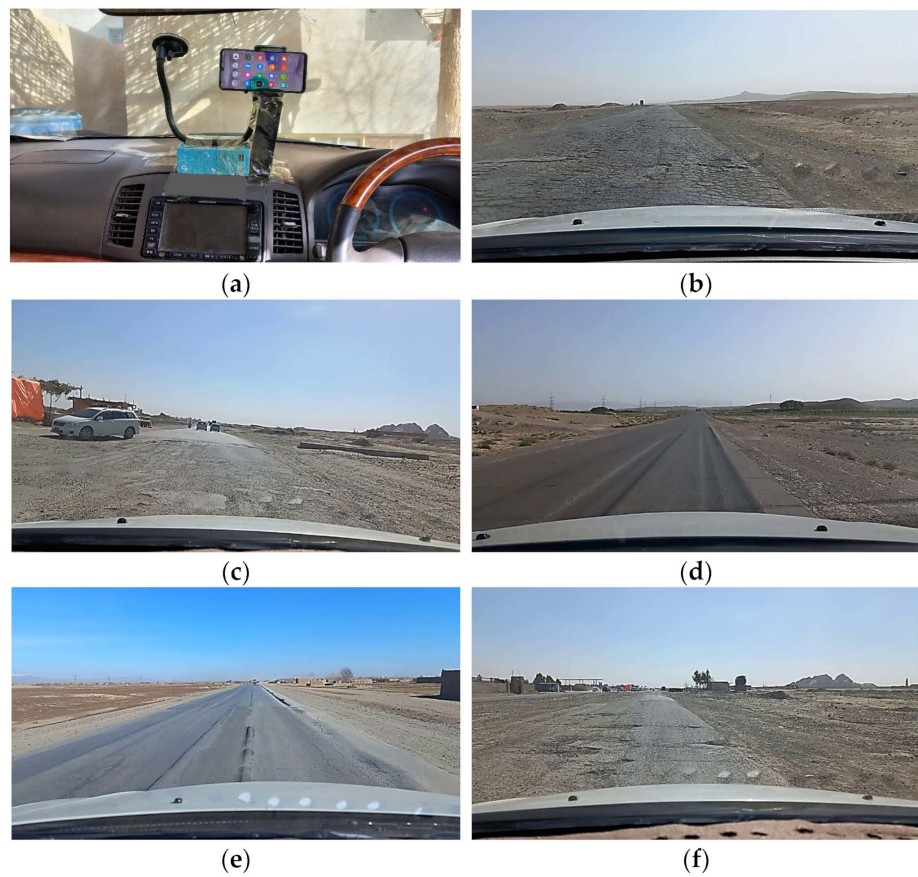

**Figure 3.** Data collection. (**a**) Smartphone setup, (**b**) alligator cracking, (**c**) raveling, (**d**) rutting, (**e**) corrugation, and (**f**) pothole.

### 3.3. Data Preparation

After data collection, data preparation was executed which consisted of checking data consistency, completeness, and accuracy. Two main errors would usually happen in the data collection process, including systematic and non-systematic errors. The former is adjustable, but the latter is inherent in any data collection. Systematic errors would occur in the data collection or evaluation process, including data collection malfunction or misclassifications of a pavement distress type. Data preparation is of significant importance as researchers have claimed that, as these errors are addressed, the number of pavement sections that need rehabilitation actions decrease by 83% and that an additional 22% of pavement sections are identified that do not require maintenance [46].

One inconsistency that could happen in the data collection process is the inexact determination of the location of a segment in different surveys over the life cycle of the pavement. There would be two reasons for this inconsistency: (1) the surveyors, data collection method, and tools might be changed in various surveys or (2) the geo-referenced tool (GPS or encoder) might be inaccurate. Both of these are inevitable, especially in developing countries. Therefore, it was decided to combine shorter segments with similar conditions

to build up a longer segment and average their pavement condition. Furthermore, outliers would be a typical issue, due to missing data or inaccurate values. For instance, if pavement condition is improved over time without maintenance or rehabilitation actions, this could be a sign of unreasonable data. This unusual improvement could occur due to either missing data, i.e., treatment was executed but not reported, or inaccurate values, i.e., tools or expert errors in objective (distress density) or subjective (distress type and severity) indication, respectively. The outliers can be easily detected by a trigger level (e.g., three standard deviations from the mean) and removed from modeling. Data preparation actions conducted in this research are as follows:

- The subsection length with minimum pavement condition variability was considered to be ten meters. Nevertheless, to tackle the problem of the inaccurate location of collected data, ten adjacent subsections were combined to build a section (100 m long) with the average condition of the subsections.
- The outliers were not detected in the dataset i.e., the PCI value is not out of the range of three times the standard deviation from the mean PCI.
- The data set did not contain the missing pavement age data. Therefore, removing the missing data was not applied in the modeling process.
- There was no need for data scaling as the order of magnitude of the variables in the model, i.e., PCI and pavement age, would not have a significant difference.

After data preparation, the entire database was again thoroughly controlled to check data consistency, completeness, and accuracy. There were no missing data, and inconsistent and inaccurate values existed in the dataset.

*3.4. Data Analysis*

For data analysis, distress type, severity, and density were determined utilizing captured images from the right of way of the pavement surface. According to ASTM 6433, 20 pavement distresses (i.e., type, density, and severity) were recognized so as to be able to estimate the PCI. According to this standard, the asphalt pavement distress types that were monitored and detected in this research included alligator cracking, bleeding, block cracking, corrugation, depression, bump and sags, lane shoulder drop-off, joint reflection cracking, longitudinal and transverse cracking, edge cracking, patching and utility cut patch, pothole, polished aggregate, raveling, weathering, rutting, shoving, railroad crossing, slippage cracking, and swells. Each distress type was labeled with a severity level including low, medium, and high. The definitions of these severity levels are provided in ASTM 6433, subjectively and objectively, along with the associated images. The density of different distress severities was defined as a ratio of distress length/area to the sample area.

In this study, the distress evaluation was performed in a semi-automated manner. First, a panel of experts determined the distress type and severity by looking at pavement images. If distress was not clear in one image, the panel could find that distress in the later or former images due to the image overlap. For instance, one expert distinguished an alligator crack with medium severity. Then, that expert employed software to measure the distress density (e.g., the area of the alligator crack). For this purpose, the images were inserted in software which is able to rigorously indicate the dimension of associated distresses. The software had been calibrated before image distress quantification with a premeasured $2 \times 2$ (m) square pattern to make sure that the measurements were accurate.

The data quality can be checked in terms of certain tasks: (1) the calibration of software, (2) the provision of a standard with which the experts evaluate the pavement, and (3) a cross-random pavement sample check (comparison of image and field inspections) [1]. All of these tasks were conducted in this research to ensure data quality. First, the software calibration was carried out as explained above. Second, ASTM 6433 was applied by the experts as a standard by which to detect and measure pavement distress. Third, some sample sections were selected to compare defects detected via images versus field observations. It should be noted that the accuracy of detecting pavement distresses through pavement images was almost the same as field inspections. Thus, it was anticipated that the level of

reliability of pavement defect detection was not significantly different between image and field distress detection.

Although the automated pavement distress type and severity detection could have been extracted automatically for some pavement distresses via computer vision techniques, such techniques were not applied herein as they would not fit the research objective of this research. This research aimed to develop a pavement performance model based on the PCI. To compute PCI, all 20 asphalt pavement distresses should be detected and measured. In spite of the fact that few simple-pattern distresses, such as alligator, block, transverse and longitudinal cracking and potholes, have been already detected via deep learning [47], some of the complicated pattern pavement distress, such as bumps and sags, depression, and corrugation, have not been recognized in a fully automated approach via computer vision techniques. Thus, it would not be feasible to compute the PCI with full automation. Therefore, we decided to detect all pavement distress (i.e., type and severity) manually in order to treat all pavement distress consistently.

Table 2 shows the distress types, severity, and density which were detected in surveyed sections. The last column in this table depicts the sum of the length/area of associated pavement distress at a specific severity level. The green, yellow, and red colors illustrate the magnitude of the associated distress, i.e., the higher values become red, while the medium and lower values appear in yellow and green, respectively. Figure 4 illuminates the density of distresses captured on the surveyed sections. This figure shows that weathering and rutting are the major distresses in the monitored sections. It also shows that 46%, 17%, and 37% of distress severities are high, medium, and low, respectively.

**Table 2.** Distress type, severity, and density.

| Distress Type | Severity | Density (sqm or m) | | | | |
| --- | --- | --- | --- | --- | --- | --- |
| | | Mean | Std | Min | Max | Sum |
| Alligator Cracking | Low | 1311 | 1869 | 0 | 6108 | 20,969 |
| | Medium | 1383 | 1383 | 59 | 5983 | 22,121 |
| | High | 552 | 552 | 25 | 2364 | 8835 |
| Bleeding | Low | 6 | 6 | 0 | 47 | 96 |
| | Medium | 5 | 5 | 0 | 27 | 77 |
| | High | 2 | 2 | 0 | 26 | 26 |
| Block Cracking | Low | 21 | 21 | 0 | 236 | 338 |
| | Medium | 157 | 157 | 0 | 1604 | 2507 |
| | High | 34 | 34 | 0 | 325 | 551 |
| Corrugation | Low | 0 | 0 | 0 | 0 | 0 |
| | Medium | 0 | 0 | 0 | 0 | 0 |
| | High | 0 | 0 | 0 | 0 | 0 |
| Depression | Low | 0 | 0 | 0 | 0 | 0 |
| | Medium | 5 | 5 | 0 | 39 | 78 |
| | High | 0 | 0 | 0 | 0 | 0 |
| Bumps and Sags | Low | 196 | 196 | 0 | 1537 | 3139 |
| | Medium | 269 | 269 | 0 | 1517 | 4298 |
| | High | 284 | 284 | 0 | 1863 | 4538 |
| Lane/Shoulder Drop-off | Low | 0 | 0 | 0 | 0 | 0 |
| | Medium | 3 | 3 | 0 | 40 | 40 |
| | High | 0 | 0 | 0 | 0 | 0 |
| Joint Reflection Cracking | Low | 0 | 0 | 0 | 0 | 0 |
| | Medium | 0 | 0 | 0 | 0 | 0 |
| | High | 9 | 9 | 0 | 92 | 149 |
| Longitudinal & Transverse Cracking | Low | 429 | 429 | 0 | 1671 | 6871 |
| | Medium | 1383 | 1383 | 41 | 5348 | 22,126 |
| | High | 258 | 258 | 0 | 792 | 4121 |
| Edge Cracking | Low | 80 | 80 | 0 | 654 | 1277 |
| | Medium | 67 | 67 | 0 | 405 | 1070 |
| | High | 129 | 129 | 0 | 628 | 2064 |

**Table 2.** *Cont.*

| Distress Type | Severity | Density (sqm or m) | | | | |
|---|---|---|---|---|---|---|
| | | Mean | Std | Min | Max | Sum |
| Patching & Utility Cut Patch | Low | 122 | 122 | 0 | 1002 | 1945 |
| | Medium | 7 | 7 | 0 | 38 | 119 |
| | High | 1 | 1 | 0 | 20 | 20 |
| Potholes | Low | 0 | 0 | 0 | 0 | 0 |
| | Medium | 3 | 3 | 0 | 50 | 50 |
| | High | 1738 | 1738 | 0 | 8240 | 27,800 |
| Polished Aggregate | Low | 36 | 36 | 0 | 210 | 580 |
| | Medium | 81 | 81 | 0 | 400 | 1300 |
| | High | 65 | 65 | 0 | 270 | 1032 |
| Ravelling | Low | 13 | 13 | 0 | 140 | 200 |
| | Medium | 3 | 3 | 0 | 35 | 49 |
| | High | 3837 | 3837 | 0 | 53,066 | 61,391 |
| Weathering (Surface Wear) | Low | 17,097 | 17,097 | 0 | 147,669 | 273,558 |
| | Medium | 2362 | 2362 | 685 | 4765 | 37,798 |
| | High | 912 | 912 | 26 | 3795 | 14,588 |
| Rutting | Low | 1321 | 1321 | 0 | 5590 | 21,138 |
| | Medium | 2723 | 2723 | 0 | 10,619 | 43,560 |
| | High | 16,165 | 16,165 | 0 | 137,726 | 258,641 |
| Shoving | Low | 555 | 555 | 0 | 4100 | 8887 |
| | Medium | 802 | 802 | 0 | 3724 | 12,837 |
| | High | 446 | 446 | 0 | 2772 | 7142 |
| Railroad Crossing | Low | 0 | 0 | 0 | 0 | 0 |
| | Medium | 0 | 0 | 0 | 0 | 0 |
| | High | 0 | 0 | 0 | 0 | 0 |
| Slippage Cracking | Low | 0 | 0 | 0 | 0 | 0 |
| | Medium | 0 | 0 | 0 | 8 | 8 |
| | High | 0 | 0 | 0 | 0 | 0 |
| Swell | Low | 0 | 0 | 0 | 0 | 0 |
| | Medium | 0 | 0 | 0 | 0 | 0 |
| | High | 0 | 0 | 0 | 6 | 6 |

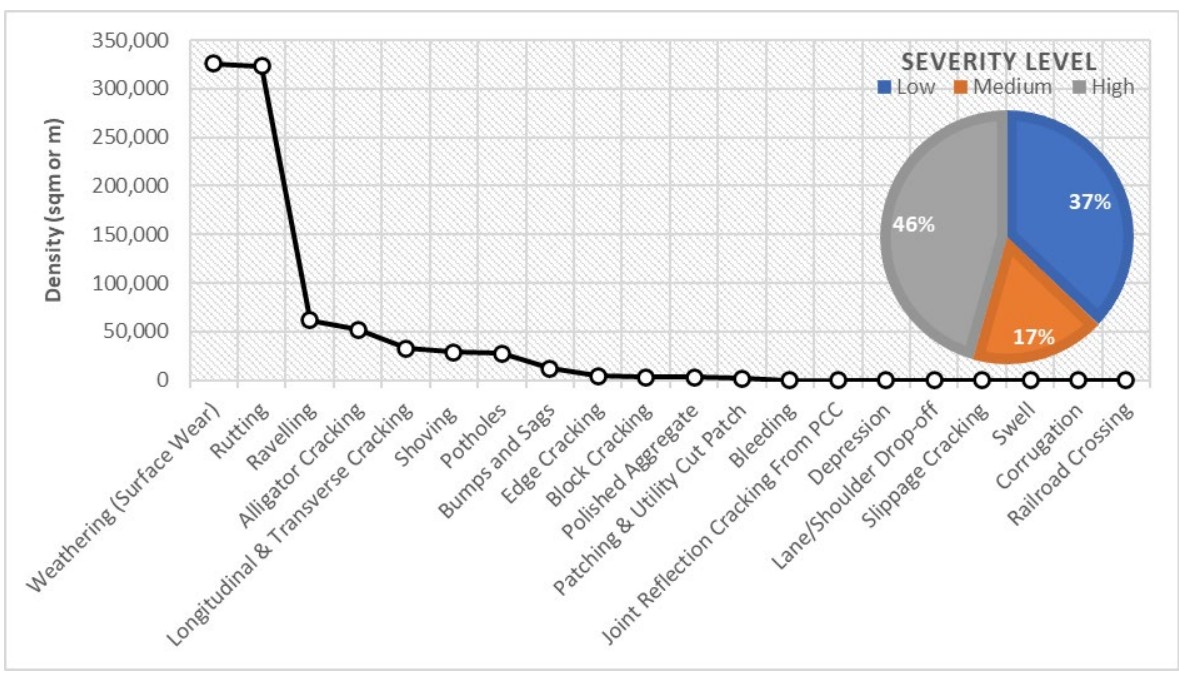

**Figure 4.** Pavement distress density of the monitored sections.

Ultimately, using distress type, severity, and density, the PCI was calculated according to ASTM 6433 via a written code in MATLAB. Table 3 depicts the average and standard deviation of the PCI and pavement age for different sections along with their length and family number (obtained from the experts from the Road Ministry of Afghanistan). Table 3 clearly shows that, as pavement age increases, the average PCI decreases, and the standard deviation of the PCI increases. In the last two columns of this table, the lower values of the average PCI and the higher values of the PCI standard deviation become red, while the higher value of the average PCI and the lower value of the PCI standard deviation appear green. The medium values are shown in yellow and orange. Figure 5 elaborates that the highest number of sections' PCI was in the "Fair" condition, in the range between 40 and 55 according to the following PCI condition ranges: Good (100–85), Satisfactory (85–70), Fair (70–55), Poor (55–40), Very Poor (40–25), Serious (25–10), and Failed (10–0) [45].

**Table 3.** PCI and pavement specifications for different surveyed sections.

| Section ID | Start | End | Length (km) | Family # | # of Section | Major Distress | Pavement Age (yr) | PCI_Avg | PCI_Std |
|---|---|---|---|---|---|---|---|---|---|
| NH01 | Gelan | Moqor | 21.7 | 1 | 217 | Weathering | 19.6 | 48.56 | 20.76 |
| NH02 | Shahjoy | Gelan | 38.6 | 1 | 386 | Weathering | 19.6 | 31.01 | 19.33 |
| NH03 | Moqor | Shamali | 34 | 2 | 340 | Weathering | 19.7 | 56.54 | 21.00 |
| NH04 | Shamali | Qarabagh | 36.2 | 2 | 362 | Aligator Crack | 19.9 | 60.35 | 28.23 |
| NH05 | Deh Tut | Kaj Ab | 32.9 | 3 | 329 | Weathering | 15.8 | 82.74 | 16.38 |
| NH06 | Kaj Ab | Farah | 22.6 | 3 | 226 | Weathering | 15.6 | 86.37 | 14.41 |
| NH07 | Washer | Delaram | 33.6 | 4 | 336 | Weathering | 16.6 | 83.42 | 13.56 |
| NH08 | Karwangah | Washer | 33.7 | 4 | 337 | Weathering | 16.8 | 86.73 | 13.51 |
| NH09 | Qarabagh | Sufra | 32.1 | 5 | 321 | Long & Tran Crack | 20.0 | 47.11 | 17.11 |
| NH10 | Sufra | Ghazni | 36.2 | 5 | 362 | Long & Tran Crack | 20.0 | 66.54 | 23.85 |
| NH11 | Gereshk | Shorawak | 33.6 | 6 | 336 | Rutting | 17.1 | 77.41 | 20.24 |
| NH12 | Shorawak | Karwangah | 33.6 | 6 | 336 | Rutting | 17.0 | 85.32 | 16.29 |
| NH13 | Herat | Shakiban | 45.1 | 7 | 451 | Rutting | 7.0 | 90.76 | 8.53 |
| NH14 | Shakiban | Islam Qala | 54.6 | 7 | 546 | Weathering | 6.7 | 94.17 | 7.07 |
| NH15 | Delaram | Golistan | 33.8 | 8 | 338 | Rutting | 16.0 | 84.72 | 14.48 |
| NH16 | Golistan | Deh Tut | 38.5 | 8 | 385 | Raveling | 15.9 | 85.38 | 11.07 |

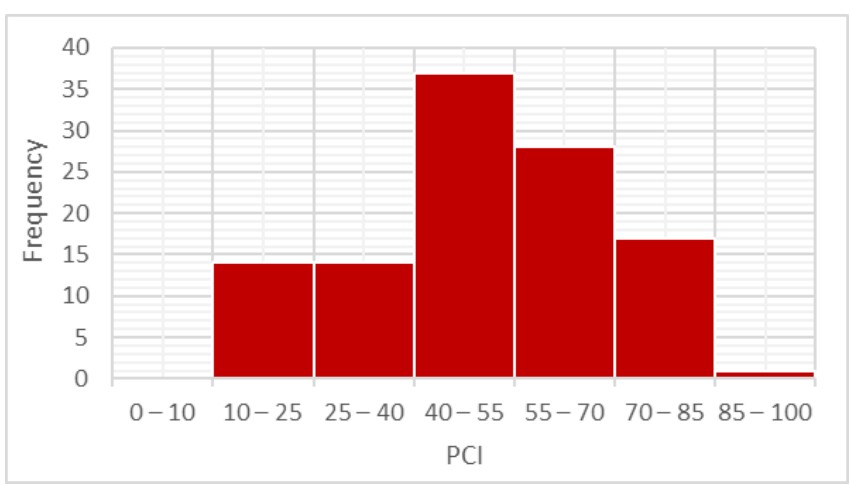

**Figure 5.** Pavement condition index for the entire collected data.

*3.5. Data Modeling*

3.5.1. Model Development

The pavement condition of a transportation network changes widely across a geographic area. This spatial variation is due to variability in the pavement structure, traffic load, and weather conditions of pavement over the network. This variability can be addressed by homogenous segmentation, i.e., by dividing the network into segments with almost consistent pavement conditions.

Comprehensive pavement historical data were not available for the Afghanistan road network, i.e., the exact objective amount of pavement criteria encompassing pavement structure and layer thickness, weather conditions, and traffic loading of different segments were unavailable. Therefore, each one of these criteria was subjectively divided into two levels to be assigned to each family. For this purpose, pavement structure, weather conditions, and traffic loading were divided into thick (pavement thickness $\geq$ 150 mm) and thin (pavement thickness < 150 mm), harsh (annual freeze-thaw cycles $\geq$ 15), and mild (annual freeze-thaw cycles < 15), and heavy (AADT $\geq$ 12,000) and low (AADT < 12,000), respectively. As each one of these three criteria had two levels, the full factorial experimental design, comprising a total of $2^3$ (i.e., 8) families, was defined and is presented in Table 4 for performance model development. The number of levels for each criterion could have been more than two, e.g., defining three groups for traffic load (i.e., heavy, medium, and low) instead of two. However, for two reasons, two levels for each criterion were selected: (1) lack of data made it hard to find enough samples for each family to build up a model and (2) no significant difference would be distinguished between performance models of the extra created families. It could be anticipated, from an engineering sense, that Family 2, with a thick pavement structure, low traffic load, and mild weather conditions, would perform better than Family 7, with a thin pavement structure, high traffic load, and harsh weather conditions. This would be confirmed later via the pavement performance model development.

**Table 4.** Pavement family characteristics.

| Family Number | Pavement Structure | Traffic Load | Weather Condition |
|---|---|---|---|
| 1 | Thick | Low | Harsh |
| 2 | Thick | Low | Mild |
| 3 | Thick | Heavy | Harsh |
| 4 | Thick | Heavy | Mild |
| 5 | Thin | Low | Harsh |
| 6 | Thin | Low | Mild |
| 7 | Thin | Heavy | Harsh |
| 8 | Thin | Heavy | Mild |

For developing models for pavement families, a univariate regression model was employed to ensure it was easy and clear enough (not a black box, such as with meta-heuristic models) to implement in developing countries. As mentioned earlier, the total pavement length that was surveyed was 558.7 km. The total pavement length was divided into 100 m sections, resulting in 5587 sections. Due to the similarity of adjacent pavement sections in each family, almost every 5 km of adjacent sections was defined as a segment leading to 112 segments. For the sake of modeling, the dataset, including 112 segments, was randomly divided into subsets of train and test with the dataset portions of 80% and 20%, respectively. An attempt was made to randomly select samples from different pavement conditions in the train and test datasets. This sampling technique would avoid sampling bias. Sampling bias would result in signs of underfitting or overfitting. In the end, the RMSE and $R^2$ of the model were calculated and reported as the model's metrics. The error was defined as the difference between the predicted (by the model) and actual values of the PCI.

To develop a model for PCI, based on pavement age for different families, the average PCI of pavement segments was plotted against pavement age. In Figure 6, the empty blue circles show the mean PCI for each segment. The primary aim was to develop a model for each family. An attempt was made to build a model for each family, as depicted in this figure and which are represented by solid lines with filled start and end points. However, the models were not meaningful due to the limited range of pavement age of each family (mostly between 1 and 3 years) that was in turn due to the limited amount of data accessible for each family. Therefore, we decided to treat all PCI data as a single family, meaning

that only one pavement performance model was developed. Thus, herein, the average PCI of all the segments was employed to construct a performance model, the solid back curve (so-called master curve), combining all families. Generally speaking, this curve would represent the performance of the entire families; however, it missed some detailed information about each family as it was all combined in a single performance model. As more information will be collected from each family, a single comprehensive model can be developed for each family which would represent its performance more accurately.

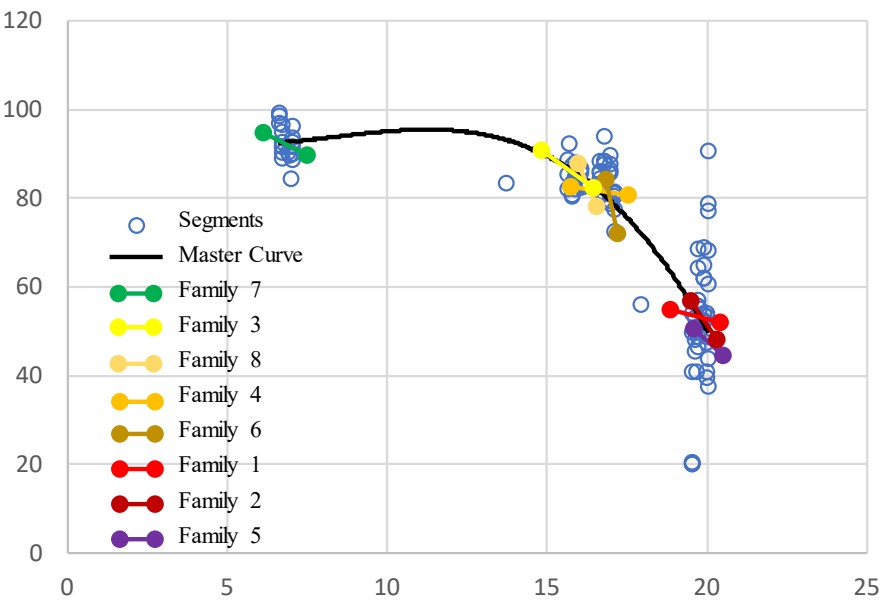

**Figure 6.** Fitted lines to families versus all together.

Using the univariate linear and non-linear regression techniques, different models, i.e., simple, second-, third-, and fourth-order polynomial, exponential, power, and logarithmic models, were built and fitted to the train data (80% of the total gathered data). The best model was shown to be the third-order polynomial model concerning model errors (RMSE), coefficient of determination ($R^2$), engineering sense (ES), related literature (RL), and underfitness/overfitness (UO) presented in Table 5. As can be seen in this table, the best model in terms of $R^2$ and RMSE is the fourth-order polynomial; however, it does not match with engineering sense (ES) and related literature (RL). It also suffers from overfitting (UO). Thus, the best-performing model after this is the third-order polynomial model, which matches engineering sense (ES), coincides with related literature (RL), and does not overfit and underfit.

**Table 5.** Model specifications.

| Type | Equation | $R^2$ | RMSE | ES | RL | UO |
|---|---|---|---|---|---|---|
| First-order polynomial | $y = -2.90x + 120.30$ | 0.4411 | 14.54 | Yes | No | Yes |
| Second-order polynomial | $y = -0.60x^2 + 12.95x + 32.25$ | 0.6993 | 10.67 | No | No | Yes |
| Third-order polynomial | $y = -0.03x^3 + 0.85x^2 - 6.31x + 106.68$ | 0.7003 | 10.65 | Yes | Yes | No |
| Fourth-order polynomial | $y = 0.0362x^4 - 2.0892x^3 + 42.96x^2 - 370.84x + 1207.7$ | 0.7189 | 10.34 | No | No | Yes |
| Exponential | $y = 143.48e^{-0.045x}$ | 0.3876 | 15.93 | No | No | No |
| Power | $y = 254.8x^{-0.474}$ | 0.3202 | 16.54 | No | No | No |
| Logarithmic | $y = -31.24ln(x) + 158.69$ | 0.3549 | 15.63 | No | No | No |

As can be seen in Figure 7a, the simple or first-order polynomial model not only expresses low fitness but also cannot represent the variation in the pavement degradation

rates. Moreover, the second-order polynomial model does not make engineering sense as it shows a significant increase/upgrade in the PCI in the initial stages which would not be feasible (in the case of not conducting maintenance actions). The third-order polynomial model, from years 7 to 11, shows a small increase of about 2%, going up from 93% to 95%. This increase, which is an error, is negligible on a scale of 100. The fourth-order polynomial model clearly expresses sharp decrease and increase, which do not make engineering sense. Regarding the pattern of data, the order one and two polynomial models suffer from underfitting, while order four indicates clear overfitting. Figure 7a also depicts that, as pavement age increases, the standard deviation of the PCI increases. Figure 7b illuminates the PCI histogram for the entire segments, which represents a slight positive skewness, in turn meaning that the bulk of the data are located at the medium and higher PCI, and that there are few very low PCI values. Moreover, Figure 7c expresses how the non-linear regression models, i.e., exponential, power, and logarithmic, fitted to the data. These did not only not fit well to the data (show high RMSE and low $R^2$) but also did not have engineering sense, as they present a sharp decrease in the initial years after construction. This does not match with reality as the PCI should have decreased slightly over primary ages.

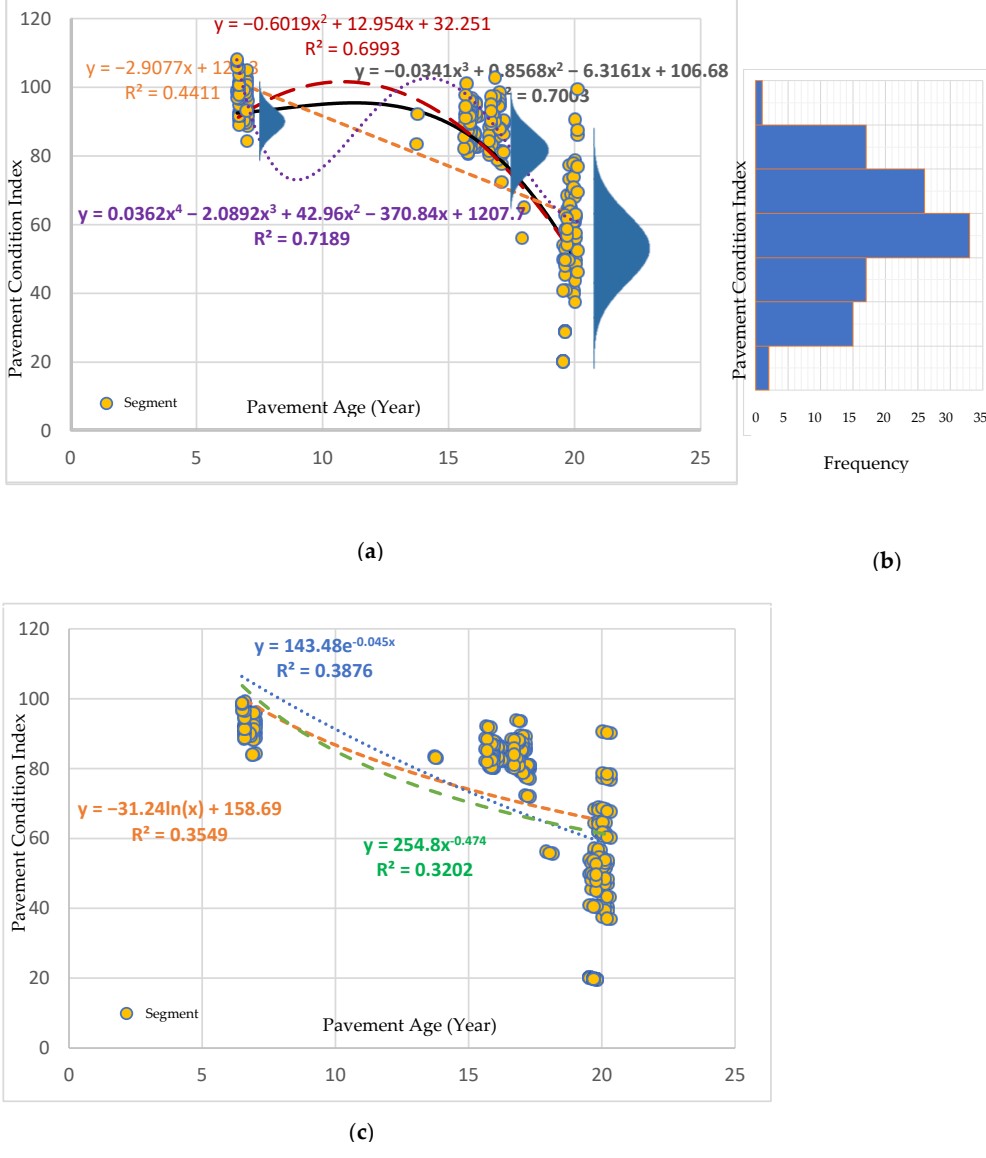

**Figure 7.** Models fitted to the data. (**a**) Linear regression models, (**b**) histogram of segments' PCI, and (**c**) non-linear regression models.

It is concluded that, as clearly shown in Table 5 and Figure 8a, the best-fitted model is the third-order polynomial model. The significant finding from this model is its deterioration rate which is higher in the pavement age range of 15 to 20 years than in initial ages (i.e., 0 to 15). This could be a vital warning for road authorities to run pavement preventive maintenance action before this range, so as to prohibit the sharp degradation in road conditions which leads to much higher corrective maintenance cost as compared with proactive maintenance.

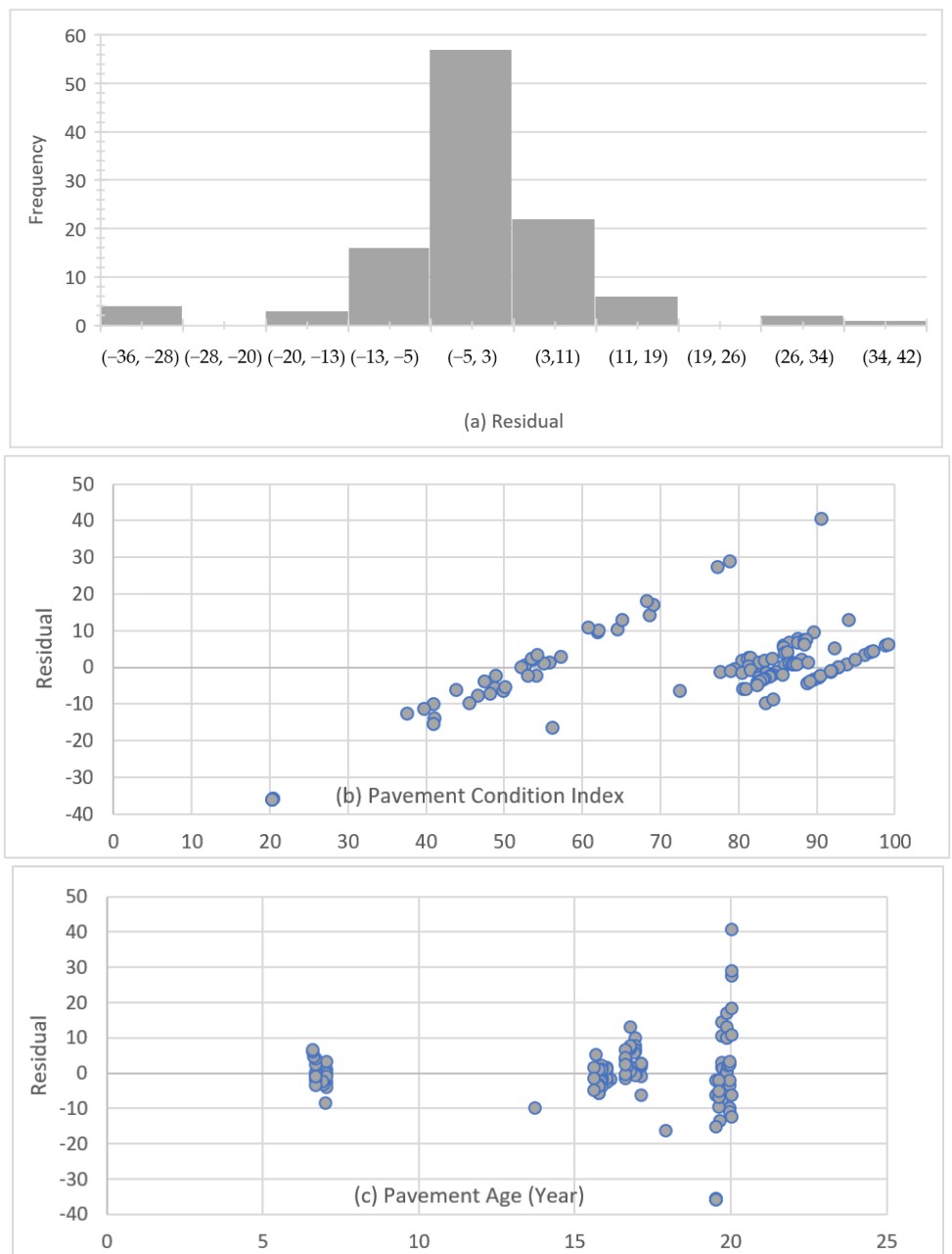

**Figure 8.** Residual diagnostics. (**a**) Histogram of model residuals, (**b**) model residual against the dependent variable, and (**c**) model residual against the independent variable.

The implications of this methodology are that, in the case of a country or city with no or limited pavement condition data, it would be possible, first, to collect data with a cost-effective and adequately accurate tool, i.e., a smartphone, and, second, to build up

a primary pavement performance model by which to represent the deterioration rate of pavement in the pavement age range of the monitored sample sections.

Such a pavement performance model may encounter some errors and would perform with a higher accuracy if these errors can be decreased. For instance, the sparse sample segments' pavement age causes errors in model development, i.e., if the sample segments' age covered a wider range and was not scattered, the model errors would be decreased.

Moreover, in the case of regular pavement condition data collection, more data would enrich the model. More data collection can be executed in three ways. Firstly, each segment can be monitored repeatedly over its lifespan. Therefore, each segment can be compared with itself over different time sections which could help in developing stage-based models such as Markov chain models. Secondly, more segments in each family can be investigated with a wider range of pavement ages, so as to be able to develop a performance model for each family. Finally, more pavement criteria can be acquired. This would, furthermore, result in the development of a more precise model if the exact pavement criteria of inventory data, traffic loading data, and weather condition data would be gathered. In such a case, these data can also be applied as independent variables in the process of modeling, to build up multiple regression models and leading to enhancement of the models' performance.

Because there have been no pavement performance models developed for Afghanistan, there is no possibility to compare the results of this study with similar previous work in the same country; however, they can be generally compared with similar attempts in other countries to ensure that the trend of the PCI deterioration over time would be approximately the same as previous work. As compared with the related literature, it was concluded that the pavement performance model that was developed in this study also perfectly expressed the same trend as other researchers have claimed [48–50]. The similarities between the developed model and the previous work are twofold. First, the pavement degradation rate is lower over the initial ages after construction than later ages. Such performance models generally degrade sharply after 75% of their life cycle, which can be clearly noticed in the developed model in this study in Figure 8a. Secondly, the most common pavement performance model curve shape in the related literature is an S shape, which is similar to the developed model herein. The S shape curve would fit best to the pavement performance model, as it can represent two different degradation rates (i.e., lower and higher). The third-order polynomial model proposed in this study exactly follows the same S shape introduced by previous researchers.

3.5.2. Model Assumptions

After developing regression performance models, regression assumptions were studied. For this purpose, the model should be scrutinized for residual diagnostics, multicollinearity, heteroscedasticity, and autocorrelation. First, for residual diagnostics, residuals/errors were plotted against dependent and independent variables. The histogram of residuals is illuminated in Figure 8a. As observed in this figure, the residuals follow a normal distribution. Additionally, the Shapiro–Wilk test approved the normality of the residuals and the model was, therefore, well-specified. Nevertheless, as can be seen in Figure 8b,c, there is evidence of a correlation that is distinguished between residuals and PCI and pavement age. This acts as a warning of heteroscedasticity that is discussed below. Second, to check the multicollinearity of the independent variables, a variation inflation factor (VIF) test and correlation matrix should be carried out. Because there was only one independent variable (i.e., pavement age) applied in this model, no multicollinearity could exist. Third, the heteroscedasticity was tested using the Goldfeld–Quandt and Breusch–Pagan tests. The former test showed a calculated F-value equal to 17.97, which is more than the critical F-value of 1.56 (at the 95% confidence level), resulting in evidence of heteroscedasticity. The later test showed a calculated $p$-value of 0.0033, which is less than the critical $p$-value of 0.05 (at the 95% confidence level), confirming the former test result. It should be noted that, as can be seen in Figure 8a, it is a common issue that, because the uncertainty is enhanced, as pavement age increases the standard deviation of PCI values

also increases. Especially in cases such as this study, where valid historical data are lacking, the reliability of data, specifically for older pavement, drastically decreases. Finally, the autocorrelation of the model was checked through the application of the Durbin Watson (DW = 0.63 showing positive autocorrelation) and autocorrelation function (AFC) tests, which both showed perfectly that the model had autocorrelation as expected because it represents the correlation of PCI over time (pavement age).

### 3.5.3. Model Validation

As explained above, the model generalization was controlled using test data. The test data, which were not used in the modeling process, were employed to compare the predicted and actual data. For this purpose, a scatter plot was drawn to show how predicted and actual PCI on the test data are coincident, as depicted in Figure 9. This coincidence can be clearly understood from this figure. Figure 9 shows that the line fitted to the data closely follows the Y = X line, with a determination coefficient of 0.97, which means the predicted PCI is almost identical to the actual PCI according to the test dataset. Moreover, a two-sample *t*-test was applied at a 95% confidence level to ensure that two subsets of data (predicted and actual) under the test dataset were not significantly different from each other, as expressed in Table 6. As elaborated in this table, the calculated *p*-value of the difference between predicted and actual PCI is 0.92, which is more than the critical *p*-value of 5%, leading to the 95% confidence level in their similarity. This is the null hypothesis, which states that the equality of the mean of the predicted and actual PCI of the test dataset cannot be rejected. Therefore, the model was perfectly validated.

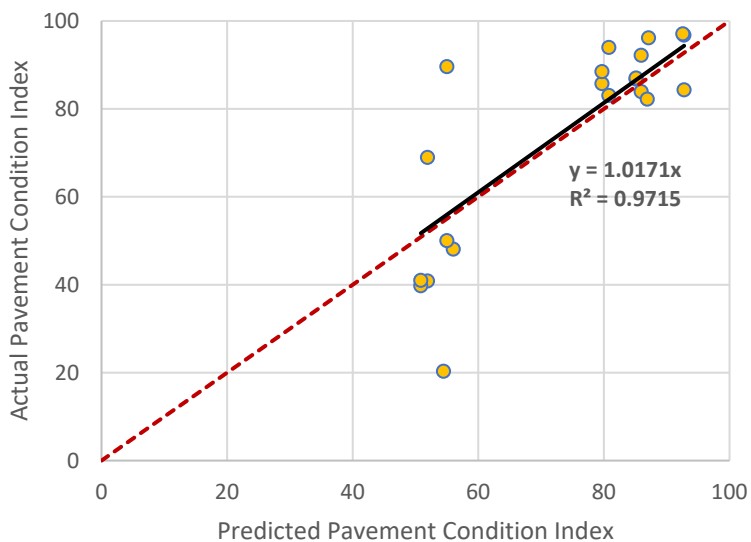

**Figure 9.** Model validation.

**Table 6.** Model validation.

| Metrics | Actual PCI | Predicted PCI |
|---|---|---|
| Mean | 73.68485 | 74.2665 |
| Variance | 530.5308 | 280.558 |
| Observations | 22 | 22 |
| Pooled variance | 405.5444 | NA |
| Hypothesized mean difference | 0 | NA |
| df | 42 | NA |
| t Stat | −0.09579 | NA |
| P (T ≤ t) two-tail | 0.924139 | NA |
| t Critical two-tail | 2.018082 | NA |

It is concluded that the model is simple to use, cost-effective to build, easy to implement, and clear to understand. It furthermore has high-performance metrics and can be

successfully validated with unseen data. Although this model was built using pavement data collected from the Afghanistan road network, it can be easily implemented in other developing countries following the same methodology as presented herein.

## 4. Conclusions

A pavement performance model is of significant importance for pavement maintenance planning, especially in a developing country such as Afghanistan, whose economy relies heavily on freight and people transit across its road network from neighboring countries. To date, there has been no performance model developed for this county. The main objective of this research was to develop a pavement performance model for Afghanistan's arterial road network which is cost-effective, easy to implement, and simple to understand, using an inexpensive data collection procedure via a smartphone. A performance model was developed and validated using a univariate regression method covering over 550 km of roads in Afghanistan expressing the PCI over the pavement age with a third-order polynomial model. The following findings have been achieved in this research study.

(1)  Pavement performance model development was not feasible for each pavement family due to the limited range of pavement age and lack of sample data.
(2)  Simple linear, polynomial, and non-linear regression models were fitted to the pavement condition data (PCI) to find the best performance. The best-performing model was the third-order polynomial model.
(3)  The third-order polynomial model's coefficient of determination and root mean squared error were 0.70 and 10.5, respectively.
(4)  The model regression assumptions were successfully checked, including uniformity of residuals, homoscedasticity, no multicollinearity, and no autocorrelation.
(5)  The model was successfully validated with unseen or test data (20% of the total dataset) via the checking of a two-sample $t$-test and a high correlation between the predicted and actual PCI.
(6)  Other developing countries with limited budgets and a lack of sophisticated automated pavement data collection tools can apply the proposed systematic approach in this research.
(7)  The limitation of this study was the lack and sparsity of sample data over the lifespan of asphalt pavement, which resulted in the development of a primary pavement performance model. The model can only predict the PCI in the range of data fed into it between 6 and 20 years.
(8)  The other limitation is that the model presents a general pavement deterioration trend over all pavement conditions, regardless of pavement criteria such as pavement structure, traffic loading, and weather conditions. The model cannot specifically predict the future condition of pavement for a region with a specific pavement criterion.
(9)  It is suggested that other indices, such as the IRI, can be captured via embedded smartphone sensors such as an accelerometer and gyroscope. The combination of PCI and IRI can be utilized for pavement maintenance planning.
(10) It is suggested that the primary model (prior probability), such as that developed in this study, can be combined with more future field investigation data, resulting in increasing model accuracy (posterior probability) via a technique such as the Bayesian model.

**Author Contributions:** Conceptualization, A.G.; methodology, A.G.; software, S.W.; validation, S.W.; formal analysis, S.W.; investigation, S.W.; resources, S.W.; data curation, S.W.; writing—original draft preparation, S.W.; writing—review and editing, A.G.; visualization, S.W.; supervision, A.G.; project administration, A.G. All authors have read and agreed to the published version of the manuscript.

**Funding:** This research received no external funding.

**Data Availability Statement:** The data presented in this study are available on request from the corresponding author. The data are not publicly available due to privacy.

**Conflicts of Interest:** The authors declare no conflict of interest.

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
