# Peer review of "Smartphone-Based Cost-Effective Pavement Performance Model Development Using a Machine Learning Technique with Limited Data"

_infrastructures, doi:10.3390/infrastructures9010009_

Round 1
Reviewer 1 Report
Comments and Suggestions for Authors
Thanks for a well thought out methodology to determine the PCI for such an extensive network. SInce this document does use image files it would have been nice to see a few photos of various levels of damage to the pavement sections studied and reported on in this paper. Perhaps an additional paper could focus on this aspect to assist others in better understanding how image files were assigned a level of damage.
Comments on the Quality of English LanguageA simple grammar check using AI or built in grammar checker should catch the minor needed revisions. This would help make this an easier read.
Author Response
We would like firstly to take this opportunity to thank the reviewer for her/his helpful and valuable suggestions. These suggestions were beneficial during the revision process and have been incorporated into the revised manuscript. The uploaded file includes two sections: (1) a point-by-point response to the reviewers' comments and (2) a highlighted paper with changes based on the comments.

Reviewer 2 Report
Comments and Suggestions for Authors
Here my comments for the Authors:
Page 1/18 - Lines 34-35
I would suggest to remark, considering the referenced study [1] is 23 years old, that .....since more than 20 years the benefits of PMS are recognized and confirmed.
Page 3/18 - Figure 1
the text is reporting the terms "underfitting" and "overfitting" while the figure 1 caption reports different terms (uderlift and overlift). It would be better to avoid confusions.
Page 3/18 - Lines 104-107
Please check the text arrangement: probably the pdf conversion created some problem.
Page 4/18 - Line 175
Reflectometer should be changed in "deflectometer"
Page 6/18 - Table 2
Introducing the Table 2, some considerations of expected results, from an engineering point of view, could be already anticipated by the Authors: it seems clear that Family #2 (thick structure, low traffic and mild weather conditions) is expected to have better performance than Family #7 (thin structure, heavy traffic and harsh wether).
Page 7/18 - Data preparation and Analysis
Could be worth to add some more comment about the methodology used for PCI evaluation, in particular regarding the accuracy od the PCI evaluation through "captured images from the right of way of the pavement surface". Did the Author performed any comparison, for some of the sample units, with PCI evaluated through visual field inspection performed by trainied operator? It could be interesting for the reader understands if the accuracy of the process used (rating of the ROW images) provide a reliability of the same order of the subjectivity evaluations performed by different trained operators. Or, alternatively, quote some relavant reference in this regard.
Page 8/18 - Line 310
Could be found another way to report the PCI range?
For instance (47-59). The actual format is confusing for the reader: [47,59] is too much similar to the bibliographic references.
The same for X-axis in Figure 6.... (where the first parenthesis is "(" not in all the ranges and the last is "]"
Page 13/18 - Lines 367-368
The quality of Figure8 should be improved. Y-axis titles have a different dimension of the fonts, Figure 8c) is not identified (where in the c)?) and Figures 8a) and b) are confusing because they seem to be part of a same graph. In particular figure 8b) has no y-axis title and it looks like there are some section with a PCI higher than 100 (when referenced to 8a), the only reference the reader can find).
Page 15/18 - Line 437
0.93.... Should be 0.92? Is it the second decimal rounded of 0.924139?
Page 16/18 - Conclusion
Maybe it could be underlined that a similar process could be useful also for developed countries where local authorities have never enough budget for a sistematic pavement evaluation process due to budget constrain.
Any further development or improvement the Authors can forecast for the study? Any future combinations with other parameters than the PCI (such us IRI, collected with the same cost-efficient methodology, where a wide set of literature is already available, also considering the imporance of roughness in pavement management tools specifically developed for developing coutries)?
Author Response
We would like firstly to take this opportunity to thank the reviewer for her/his helpful and valuable suggestions. These suggestions were beneficial during the revision process and have been incorporated into the revised manuscript. The uploaded file includes two parts: (1) a point-by-point response to the reviewers' comments and (2) a highlighted paper with changes based on the comments.

Reviewer 3 Report
Comments and Suggestions for Authors
The manuscript entitled “Smartphone-based cost-effective pavement performance model development using a machine learning technique with limited data” proposes the development a pavement performance model for Afghanistan based on data collected by a smartphone. Although the topic is interesting, in my opinion, the article present serious flaws that prevent me from approving it for publication in this journal. In this context, here are some suggestions for the article to be improved and resubmitted in this or another journal.
MAJOR COMMENTS:
1) PAPER STRUCTURE: I suggest that the manuscript be restructured. The Introduction section should provide a brief introduction to the topic, delimit the research gap, the objective, highlight the novelty of the research, and provide a brief explanation of the methodology to be used. The literature review section aims to indicate the state-of-the-art on the subject. In other words, indicate which results the most recent studies have produced on the subject. As well as substantiating the most important concepts for understanding the research. It is not possible to guarantee a solid theoretical background without broadly discussing the results of previous research on the subject. In this context, the subsections, “Pavement performance model development methods”; “Modeling techniques”; and, “Pavement performance modeling in developing countries”, should be included in the Literature Review section. Furthermore, they must clearly present the results of previous works on the topic, theoretically substantiating this research. It is not enough to just cite studies that used a certain technique, or that addressed case studies in certain countries. Finally, the “Objective and scope” subsection must be positioned within the Introduction section.
2) FIGURE 1: Authors should reflect on the relevance of including this figure in the manuscript. It covers basic information, which, in my opinion, doesn't add much to the article.
3) FIGURE 2: In line 201, the authors indicate that “Pavement network segmentation” was the second stage of the research, after the literature review. However, when looking at Figure 2, it is clear that this item is far from the literature review, that is, in the third line of the figure. I suggest that the authors revise the text so that the information in the figure corresponds to the information in the text.
4) LINE 298: Tables and Figures must be positioned close to the place where they first appear. Please check all the text.
5) Authors could improve the graphic quality of figures, graphs, and tables to give the manuscript a more professional appearance.
6) TABLES 3 and 4: Authors must clearly explain what the colors in the tables mean.
7) FIGURE 5: The authors address very little about the pathologies found in the asphalt throughout the manuscript, but they include this figure which, in my opinion, leaves the reader lost in the text.
8) FIGURE 6: I really wasn't able to interpret this graphic in a practical and assertive way.
10) FIGURES 7 and 8: In my opinion, figures 7 and 8 were included in the text with insufficient explanations about their relevance and adequacy.
11) DISCUSSION: Very little discussion was presented based on the results obtained. What are the implications of using this methodology? In which situations can it be used with a low margin of error? How to improve the developed model? How do the obtained results relate to the results of previous work? In my opinion, the authors should work to enhance the results obtained in the research.
12) CONCLUSION: This section needs to be drastically improved. The authors must, at least, present their perceptions of the obtained results, present the research limitations, and suggestions for future work.
MINOR COMMENTS:
13) LINES 33-35: “From the lens of agency and user costs, researchers declared that $30 and $250 would be economized, respectively, for one dollar spent on developing, implementing, and operating a Pavement Management System (PMS) [1]”. I suggest that the sentence be rewritten in a way that makes it easier to understand.
14) LINES 105-106: I believe there was an error in the text, as some words were partially deleted.
15) LINE 233: This line displays Table 2, but I couldn't find Table1. Please check whether the table was deleted incorrectly or whether the tables need to be renumbered.
Author Response

(The authors gave the same response as above.)

Reviewer 4 Report
Comments and Suggestions for Authors
This paper proposes a smartphone-based pavement performance model to predict pavement conditions based on the collected data. The proposed models were validated against field-collected data. I have a few comments and suggestions that may help improve the clarity and impact of the research paper:
1. The caption of Figure 1 should be underfitting and overfitting.
2. Please add the missing Table 1.
3. In the proposed approach, the smartphone is used to collect pavement images and then predict pavement conditions. There is another smartphone-based approach that normally collects vehicle vibration data and uses the data to back-calculate road surface roughness. It is suggested to discuss the other smartphone-based approach and show the advantages/disadvantages of the image-based method.
4. Can you add more details on the collected dataset and how it is split into the training dataset and testing dataset?
5. How is the distress type, severity, and density information extracted from the collected smartphone images? Assuming they are manually estimated, can we leverage some computer-vision technologies to automate this process?
6. This paper used Pavement Condition Index (PCI) as the indicator of pavement surface conditions. However, it is recognized that International Roughness Index (IRI) has better correlation to ride quality, fuel consumption, and driving safety. The effect of IRI has been well studied in recent studies, such as “Mechanistic Excess Fuel Consumption of a 3D Passenger Vehicle on Rough Pavements” and "Stochastic analysis of rolling resistance energy dissipation for a tractor-trailer model" where some vehicle dynamic models (3D car model and 3D truck model) were developed to predict road IRI and their impact on the vehicle rolling resistance. It is suggested to discuss this research in the introduction part. You can mention that your research improves the existing research by introducing a rough pavement surface model.
Author Response

(The authors gave the same response as above.)

Round 2
Reviewer 3 Report
Comments and Suggestions for Authors
----